# Efficacy of combined COVID-19 convalescent plasma with oral RNA-dependent RNA polymerase inhibitor treatment versus neutralizing monoclonal antibody therapy in COVID-19 outpatients: a multi-center, non-inferiority, open-label randomized controlled trial (PlasMab)

Taweegrit Siripongboonsitti,[1,2] Nuttakant Nontawong,[3] Kriangkrai Tawinprai,[1,2] Ornpreya Suptawiwat,[2] Kamonwan Soonklang,[4] Yong Poovorawan,[5] Nithi Mahanonda[6]

**ABSTRACT** The effectiveness of neutralizing monoclonal antibodies against emerging variants with spike protein mutations has been limited. Early high-titer convalescent plasma therapy (CPT) tends to be effective and a potential treatment option for COVID-19 outpatient treatment. An open-label, 1:1, randomized controlled trial compared the efficacy of high neutralizing titer CPT with favipiravir (CPT-FPV) versus sotrovimab. The study aimed to evaluate the non-inferiority of CPT-FPV treatment compared to sotrovimab in preventing hospitalization within 14 and 28 days. Among the 136 randomized participants, 68 received CPT-FPV, while 68 received sotrovimab. The study demonstrated that CPT-FPV was non-inferior to sotrovimab in preventing hospitalization within 14 and 28 days. No significant differences between the two treatment arms were observed in emergency room visits, oxygen supplementation, or mortality. Although there were no significant disparities in radiological, virological, or inflammatory marker outcomes, sotrovimab exhibited a more pronounced reduction in IL-6 levels during days 2–5 and lower MCP-1 levels on day 14. High neutralizing titer convalescent plasma and favipiravir demonstrated non-inferior to sotrovimab in preventing hospitalization, emergency room visit, oxygen supplementation, and mortality in mild-to-moderate COVID-19. The evidence may be a choice for treatment for new variants despite the need for further study. (The trial protocol was registered in the Thai clinical trials registry no. 20220319002.)

**IMPORTANCE** This pivotal study reveals that high neutralizing titer COVID-19 convalescent plasma therapy (CPT) combined with favipiravir (FPV) is non-inferior to sotrovimab in preventing hospitalization and severe outcomes in outpatients with mild-to-moderate COVID-19 and high-risk comorbidities. It underscores the potential of CPT-FPV as a viable alternative to neutralizing monoclonal antibodies like sotrovimab, especially amid emerging variants with spike protein mutations. The study's unique approach, comparing a monoclonal antibody with CPT, demonstrates the efficacy of early intervention using high neutralizing antibody titer CPT, even in populations with a significant proportion of elderly patients. These findings are crucial, considering the alternative treatment challenges, especially in resource-limited countries, posed by the rapidly mutating SARS-CoV-2 virus and the need for adaptable therapeutic strategies.

**KEYWORDS** convalescent plasma, sotrovimab, monoclonal antibody, mild COVID-19, SARS-CoV-2, randomized controlled trial, outpatient COVID-19, non-severe COVID-19

Address correspondence to Taweegrit Siripongboonsitti, Taweegrit.sir@cra.ac.th.

The authors declare no conflict of interest.

See the funding table on p. 14.

The COVID-19 pandemic has had a devastating impact on global health, resulting in unprecedented levels of mortality and morbidity. To combat the severity of the disease, various oral antiviral agents have been developed and proven effective in preventing severe cases of COVID-19, namely, nirmatrelvir/ritonavir and molnupiravir (1–3). In Thailand, amid the surge of the delta and omicron variants, favipiravir (FPV) has been extensively deployed as the standard treatment for mild-to-moderate cases of COVID-19, adhering to the Thai COVID-19 clinical practice guidelines. This approach was necessitated by the constrained availability of novel therapeutic options (4). A real-world study within Thailand, alongside a meta-analysis, underscored the advantages of FPV treatment, revealing notable clinical improvements, diminished rates of clinical deterioration, enhanced viral clearance, decreased dependency on supplemental oxygen therapy, and reduced mortality (5–7). However, a randomized controlled trial (RCT) conducted among obese Hispanic individuals failed to demonstrate the efficacy of FPV treatment in mild-to-moderate COVID-19 (8). Given its non-linear pharmacokinetics and auto-inhibition properties, FPV's efficacy remains a topic of controversy and perpetuates a treatment gap, especially among certain ethnic groups characterized by overweight and obesity (9, 10).

Sotrovimab (VIR-7831) was a neutralizing monoclonal antibody (NmAb), which many RCTs demonstrated preventing hospitalization in outpatient COVID-19; however, there were many gaps in NmAb treatment, including the high cost and difficulty of access (11–15). While most monoclonal antibodies lose their ability to neutralize the omicron variant in laboratory tests due to mutations in the receptor-binding motif, *in vivo*, data indicate that sotrovimab remains effective in preventing COVID-19 progression, particularly in the omicron BA.2 sublineage (16, 17). The escape spike protein mutation led to NmAb resistance, while the rapid emergence of spike protein mutations in new variants hindered NmAb efficacy for COVID-19 treatment. The short-lived utility of NmAb contributed to hesitation in developing costly alternatives.

The COVID-19 convalescent plasma (CCP) transfusion was reported to treat MERS-CoV, SARS-CoV-1, and influenza virus (18). CCP therapy acts as passive immunization with a well-safety profile, although the result of convalescent plasma therapy (CPT) in various clinical trials was mixed (19). Many negative CPT trials were conducted in hospitalized and ICU-admitted patients with severe COVID-19. The REMAP-CAP, CONCOR-1, RECOV-ERY, PLACID, and RECOVER could not demonstrate the benefit of CCP in decreasing mortality, time to hospitalized discharge, organ and respiratory support free day, intubation rate, and ICU length of stay in severe and critical COVID-19 patients (19–26). A limitation of some studies included the low and varying CCP neutralizing antibody titer and prolonged duration from onset of illness to received CCP (20, 25).

The favorable outcomes of CPT were demonstrated in those who received early high neutralizing antibody titer CPT in outpatient treatment and early course in severe COVID-19 (27–31). The CSSC-004 and CCP-Argentina demonstrated the prevention of severe COVID-19 and reduction in hospitalization, including early symptomatic high-risk patients (32, 33). Although CONV-ERT trial could not demonstrate benefits in reducing hospitalization, viral clearance, and death, methylene blue used in pathogen inactivation potentially impaired Fc-region functionality IgG (34). The SIREN-C3PO and CoV-Early trial demonstrated that high-titer CCP within 1 week of symptom onset did not prevent disease progression in outpatients with COVID-19 who were at high risk of severe disease; however, SIREN-C3PO in the CCP group included more patients with multiple risk factors, including immunosuppression (35). The meta-analysis demonstrated that the positive result of CPT depended on high neutralizing antibody titer and early COVID-19 (36). Thus, early high neutralizing titer CCP might be the treatment option, particularly in resource-limited countries, providing possibilities for treating new variants (37, 38).

This study aimed to evaluate the efficacy of combined high neutralizing antibody titer CPT with favipiravir (CPT-FPV) and evaluated the non-inferiority to sotrovimab in preventing hospitalization in mild-to-moderate COVID-19 with outpatient treatment.

## MATERIALS AND METHODS

### Trial design

The multi-center, non-inferiority, open-label, randomized controlled trial was conducted at Chulabhorn Hospital, Bangkok, and Prachathiput Hospital, Pathum Thani, Thailand. The trial followed the Good Clinical Practice guidelines and the principles of the Declaration of Helsinki. The Ethics Committee approved the study for Human Research, Chulabhorn Research Institute, no. 192/2564. The trial protocol was registered in the Thai clinical trials registry no. 20220319002.

### Participants

A total of 136 eligible participants were included in the study. To be eligible, participants had to be Thai individuals over 20 years old and have a mild-to-moderate COVID-19 diagnosis confirmed by PCR testing and a body weight of 40 kg or more. Participants were required to have at least one risk factor associated with the potential deterioration to severe illness, including age over 60 years, obesity or body mass index (BMI) of 30 kg/m$^2$ or more, diabetes mellitus, chronic kidney disease, heart disease, cardiovascular disease, COPD/asthma, hypertension, liver disease with a Child–Pugh score of A–B, receiving immunosuppressive therapy, cerebrovascular accident/stroke, or active cancer. All participants experienced respiratory symptoms within 7 days of illness onset, such as fever, cough, runny nose, sore throat, dyspnea, chest discomfort, and diarrhea.

Pregnant or breastfeeding individuals, those with severe COVID-19 requiring oxygen therapy or hospitalization, individuals on anti-SARS-CoV-2 therapy for more than 24 hours, those with chronic liver disease Child–Pugh score C or end-stage renal disease or advanced-stage cancer, those who had received neutralizing monoclonal antibody or plasma within the past 60 days, or those who had a history of severe allergic reaction to FPV or sotrovimab or severe transfusion reaction were excluded from the study (Fig. 1).

### COVID-19 convalescent plasma preparation

CCP-specific donors were carefully selected based on specific criteria. Donors were chosen from individuals with a confirmed SARS-CoV-2 infection through PCR testing within the past 28 days. Male donors weighing more than 50 kg were preferred to ensure an adequate volume of plasma (>250 mL) and eliminate the possibility of pregnancy. The selected donors were aged between 18 and 60 years and had recovered from respiratory tract infection symptoms. All eligible donors met the plasma donation criteria. The neutralizing antibody titer was assessed using a micro-neutralization assay conducted at Mahidol University's Faculty of Sciences. All recruited donors had a neutralizing antibody titer of ≥1:320, exceeding the Food Drug Administration–recommended threshold of ≥1:160 and anti-SARS-CoV-2 ELISA IgG EUROIMMUN ratio ≥3.5. This titer correlated with an anti-spike RBD ELISA IgG titer of ≥1:1,350 U/mL, indicating the presence of high levels of anti-SARS-CoV-2 antibodies (31, 39, 40). Donor eligibility was further evaluated based on medical history in the past 12 months and a physical examination relevant to transfusion-transmitted infection risk. Additionally, the interval between plasma donations was required to be at least 8 weeks if the donor had previously donated a unit of whole blood or a single unit of red blood cells by apheresis. Human leukocyte antigen antibodies were screened, and negative results were obtained. The collection of CCP was conducted through plasmapheresis and processed at the national blood center of the Thai Red Cross Society. All collected CCP underwent screening for transfusion-transmitted infections, including HIV, hepatitis B and C, syphilis, and malaria. Furthermore, pathogen inactivation procedures were employed, and the stored CCP was kept at −20°C for less than 1 year.

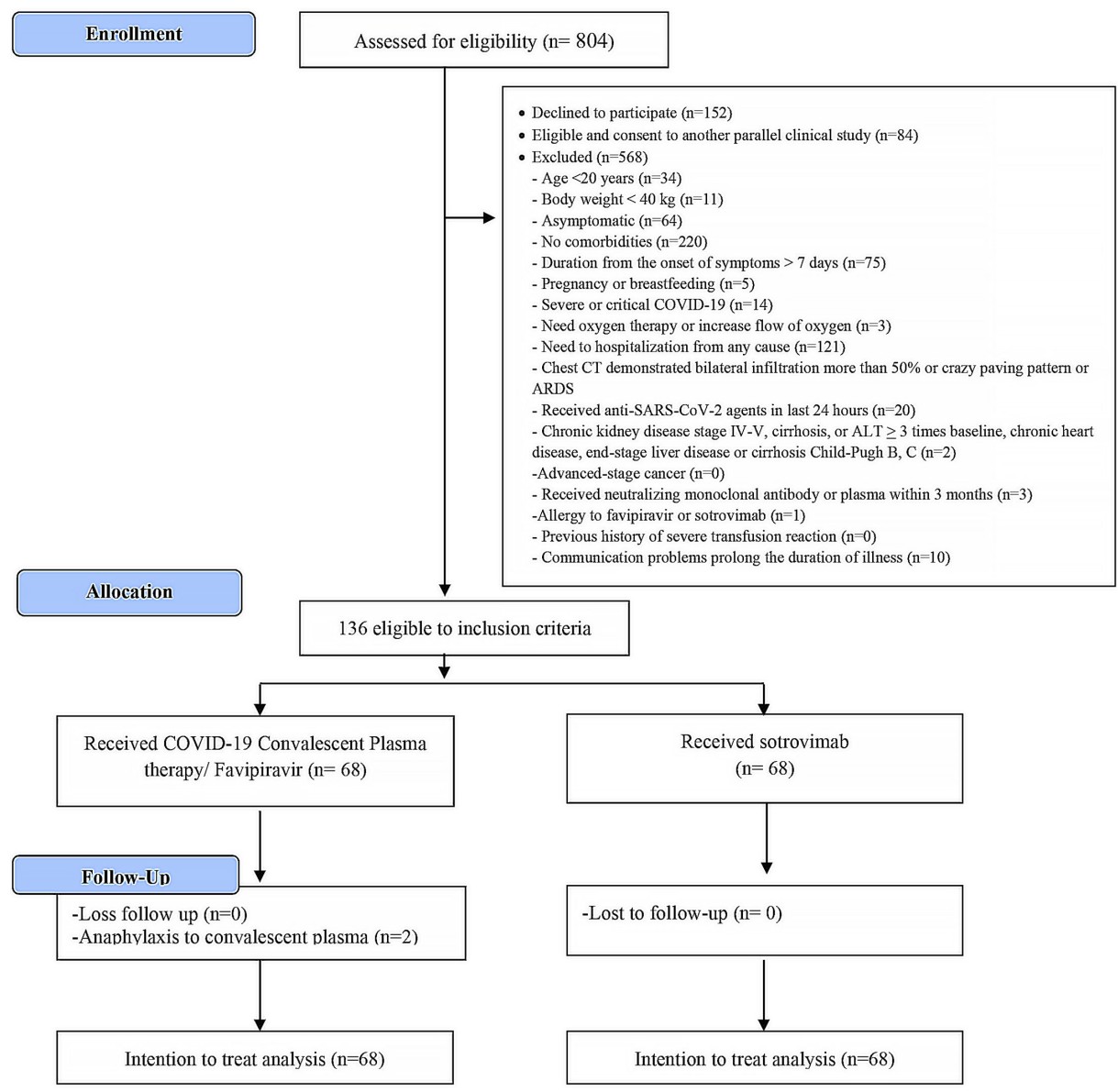

**FIG 1** Flow of enrollment and allocation.

## Interventions

Of the 136 randomized enrolled participants, 68 received a single dose of CPT and 5-day of FPV treatment, while 68 received a sotrovimab single dose. In the CPT arm, ABO-compatible CCP was transfused to patients, with a volume of 250 mL for <60 kg recipients and 300 mL for >60 kg recipients. The infusion was completed within 2 hours, with close monitoring for transfusion-related reactions and adverse events, promptly treating any observed complications. The patients received 3,600 mg of FPV loading dose on day 0 and 1,600 mg per day of FPV from days 1–4. In a patient with obesity, the FPV loading dose was 4,800 mg on day 0, then 2,000 mg per day of FPV on days later. In the sotrovimab arm, the participants received a single dose of 500 mg of intravenous sotrovimab within 1 hour. The participants received clinical, laboratory, virologic, and radiologic evaluations on days 2, 5, 14, and 28–42 and were evaluated at the outpatient clinic.

## Outcomes

The primary outcome of this study was to evaluate the efficacy of CPT-FPV in preventing hospitalization within 14 and 28 days while demonstrating its non-inferiority to sotrovimab in treating mild-to-moderate COVID-19 cases in outpatient settings. The secondary outcomes included assessing the rates of oxygen supplementation, emergency department visits within 14 and 28 days, utilization of invasive mechanical ventilation, ICU admission, mortality, clinical improvement, non-clinical progression, and changes in inflammatory biomarkers on days 2, 5, and 14. Additionally, virologic, radiologic, and cytokine measurements were taken on days 0, 5, and 14, and any observed adverse effects were recorded.

## Randomization

One hundred thirty-six participants meeting the inclusion criteria were randomly assigned in a 1:1 ratio to receive either CPT-FPV or sotrovimab using computer-based block randomization. The random allocation sequence numbers were concealed in closed envelopes held by a statistician. Non-trial-affiliated nurses conducted the enrollment, and intervention assignments were performed by pharmacists using pre-prepared sequentially numbered containers.

## Sample size

The sample size for this study was determined based on real-world data evaluating the effectiveness of sotrovimab in preventing hospitalization in cases of mild-to-moderate COVID-19. The trial results indicated that sotrovimab reduced hospitalization rates to 1% compared to 11.4% in the standard care group (41). To establish the non-inferiority and ensure that CPT-FPV had a significant difference of less than 2% compared to sotrovimab, an accepted success rate of 99% in preventing hospitalization within 28 days was set for sotrovimab, while CPT-FPV needed to achieve a success rate of 88.6%. To maintain a type 1 error of 0.05 and a statistical power of 90% for detecting non-hospitalization as a binary outcome, the study aimed to enroll 136 patients, with an additional 10% included to account for potential dropouts.

## Statistical methods

The study compared the non-inferiority of CPT-FPV to sotrovimab in preventing hospitalization. Normality was assessed using the Shapiro–Wilk test. The intention to treat (ITT) analysis employed Fisher's exact, Mann–Whitney $U$, and chi-squared tests. Categorical variables were analyzed with Pearson chi-squared and Fisher's exact test, while continuous data used an independent $t$-test for parametric data and Mann–Whitney $U$ test for non-parametric data. Logistic regression and multi-level regression models were used for specific analyses. Inflammatory markers and cycle threshold (Ct) values were analyzed using the Wilcoxon signed-rank and Mann–Whitney $U$ tests. Statistical significance was set at a two-sided $P$-value less than 0.05. STATA version 16.1 (StataCorp, College Station, TX, USA) was used for all analyses.

## RESULTS

### Trial participants

From 22 January 2022 to 23 April 2022, 804 participants who visited the COVID-19 clinic were assessed for eligibility. Of these, 152 participants declined to participate, 84 consented to another parallel clinical study, and 568 did not meet the inclusion criteria and were excluded. Ultimately, 136 participants who met the inclusion criteria were randomized, with 68 receiving CCP-FPV and 68 receiving sotrovimab. No participants were lost to follow-up throughout the study period (Fig. 1).

## Baseline characteristics

There was no difference in participants between the CCP and sotrovimab groups in age, gender, variant of concern, body mass index, duration from the onset of symptoms to treatment, vaccination status, pneumonia, cycle threshold value from PCR, and inflammatory markers. Almost all high-risk comorbidities were not different in both groups; however, hypertension was predominated at 61.8% in CPT and 42.7% in the sotrovimab group.

The age of more than 60 years was 66.1% in CPT and 69.8% in the sotrovimab arm. The most common comorbidities were diabetes, obesity, and malignancy. The omicron BA.2 sublineage was predominated in both groups, despite 60% unknown variants in both arms. The mean duration from the onset of symptoms to randomization was 2 days. Unvaccinated presented 15% vs 25% in CPT and sotrovimab, respectively. Pneumonia was presented in 30% of both arms (Table 1; Table S1).

## Outcomes

### The primary outcome

The ITT analysis was conducted to evaluate and determine that CPT-FPV exhibited non-inferiority compared to sotrovimab in terms of preventing hospitalization within 14 and 28 days [$P = 0.496$, risk difference 0.03, 95% CI (−0.01, 0.07)]. Remarkably, no hospitalization cases due to severe COVID-19 were observed in either treatment group within 28 days ($P = 1.000$) (Fig. 2; Table 2).

## The secondary outcome

### Clinical outcome

No significant differences were observed between the CPT-FPV and sotrovimab groups regarding emergency department visits within 14 and 28 days, rates of ICU admission, oxygen supplementation, invasive mechanical ventilation, and mortality.

Furthermore, the two treatment groups had no notable clinical improvement, deterioration, or stability variations. The clinical outcomes indicated that 46 (67.7%) patients in the CPT-FPV group and 44 (64.7%) patients in the sotrovimab group experienced clinical improvement by day 2, while 63 (92.7%) patients in the CPT-FPV group and 60 (88.2%) patients in the sotrovimab group exhibited clinical improvement by day 5. According to days 5 and 14, all patients had a WHO clinical progression scale (WHO-CPS) score of 2 or lower. On day 5, 85.3% of CPT-FPV patients and 79.4% of sotrovimab patients had a WHO-CPS score of 1, while 76.5% of CPT-FPV patients and 88.2% of sotrovimab patients had a WHO-CPS score of 0 by day 14 (Table 2).

In a *post hoc* subgroup analysis, no significant differences were found in the clinical outcomes among unvaccinated individuals, those who received at least a two-dose vaccination, and those with a positive RBD spike IgG antibody level greater than 1,000 (Table S11).

### Virological outcomes

No significant difference in viral clearance was observed between the CPT-FPV and sotrovimab groups. On day 5, the viral clearance rates were 7% for CPT-FPV and 13.4% for sotrovimab ($P = 0.273$). By day 14, the viral clearance rates were 83.8% for CPT-FPV and 91.2% for sotrovimab ($P = 0.195$) (Table 3; Fig. S8)

### Radiologic outcomes

The CPT-FPV group showed no significant difference in the progression and resolution of pneumonia on day 5, as evaluated by chest CT ($P = 0.258$). However, the CT severity index scores increased in the CPT-FPV group between day 0 and day 5 ($P = 0.039$) (Table 3; Fig. S4).

**TABLE 1** Baseline characteristics of participants[e,f]

| Parameters | COVID-19 convalescent plasma/favipiravir (*n* = 68) | Sotrovimab (*n* = 68) | *P*-value |
|---|---|---|---|
| Male, *n* (%) | 29 (42.65) | 29 (42.65) | 1.000[a] |
| Age, mean ± SD | 61.97 ± 1.79 | 64.34 ± 1.49 | 0.312[c] |
| Age >60 years old | 45 (66.18) | 95 (69.85) | 0.350[a] |
| Duration from the onset of illness, mean ± SD | 2.62 ± 1.46 | 2.66 ± 1.61 | 0.986[d] |
| BMI, mean ± SD | 27.47 ± 7.90 | 25.86 ± 4.67 | 0.451[d] |
| BMI, *n* (%) | | | 0.467[b] |
| 18–24.9 | 31 (45.59) | 33 (49.25) | |
| 25–29.9 | 18 (26.47) | 22 (32.84) | |
| ≥30 | 17 (25.00) | 11 (16.42) | |
| Variant (variant of concern), *n* (%) | | | 0.881[b] |
| Omicron BA.1 | 1 (1.47) | 1 (1.47) | |
| Omicron BA.1.1 | 5 (7.35) | 2 (2.94) | |
| Omicron BA.2 | 19 (27.94) | 20 (29.41) | |
| Unknown | 41 (60.29) | 41 (60.29) | |
| Previous COVID-19 infection, *n* (%) | 4 (5.88) | 2 (3.13) | 0.681[b] |
| Vaccination status, *n* (%) | | | 0.274[b] |
| Unvaccinated | 10 (14.71) | 17 (25.00) | |
| Incomplete vaccination | 0 (0.00) | 1 (1.47) | |
| Fully inactivated or ChAdOx-1 nCoV vaccination | 13 (19.12) | 12 (17.65) | |
| Third-dose vaccination | 39 (57.35) | 29 (42.65) | |
| Fourth-dose vaccination | 6 (8.82) | 9 (13.24) | |
| Comorbidities, *n* (%) | | | |
| Obesity | 18 (26.47) | 11 (16.42) | 0.155[a] |
| Malignancy | 6 (8.82) | 7 (10.29) | 0.771[a] |
| Diabetes mellitus | 17 (25.00) | 15 (22.06) | 0.686[a] |
| Hypertension | 42 (61.76) | 29 (42.65) | 0.026[a] |
| Chronic kidney disease | 1 (1.47) | 3 (4.41) | 0.619[b] |
| Coronary artery disease | 1 (1.47) | 4 (5.88) | 0.366[b] |
| Chronic lung disease | 1 (1.47) | 1 (1.47) | 1.000[b] |
| Other cardiac abnormalities | 6 (8.82) | 5 (7.35) | 1.000[a] |
| Cerebrovascular disease | 5 (7.35) | 3 (4.41) | 0.718[b] |
| Body temperature | 36.53 ± 0.42 | 36.42 ± 0.27 | 0.185[d] |
| Symptoms, *n* (%) | | | |
| Cough | 48 (72.73) | 47 (69.12) | 0.646[a] |
| Fever | 14 (21.21) | 12 (17.65) | 0.602[a] |
| Sore throat | 30 (45.45) | 27 (39.71) | 0.501[a] |
| Runny nose | 33 (50.00) | 25 (36.76) | 0.122[a] |
| Dyspnea | 1 (1.52) | 2 (2.94) | 1.000[b] |
| Chest tightness | 1 (1.54) | 3 (4.41) | 0.620[b] |
| Diarrhea | 1 (1.52) | 3 (4.41) | 0.619[b] |
| Mild illness | 47 (69.12) | 46 (67.65) | 0.854[a] |
| Moderate illness (pneumonia) | 21 (30.88) | 22 (32.35) | |
| Ct value of SARS-CoV-2 PCR, mean ± SD | 20.82 ± 4.73 | 20.08 ± 3.18 | 0.958[d] |
| Positive anti-spike RBD IgG antibodies | 54.00 (79.41) | 61.00 (89.71) | 0.097[a] |

[a]Pearson chi-squared test.
[b]Fisher's exact test.
[c]Independent *t*-test.
[d]Mann–Whitney *U* test.
[e]Categorical data are depicted as the number and percentage of participants. Continuous data are displayed as the mean ± standard deviation.
[f]*n*: number; Ct: cycle threshold; BMI: body mass index; PCR: polymerase chain reaction; SD: standard deviation; RBD: receptor-binding domain; SARS-CoV 2: severe acute respiratory syndrome coronavirus-2; COVID-19: coronavirus disease 2019; Ig: immunoglobulin.

## Inflammatory markers

There were no significant differences observed in the levels of inflammatory markers, including high-sensitive C-reactive protein (hs-CRP), D-dimer, presepsin, procalcitonin, ferritin, and erythrocyte sedimentation rate, on days 0, 2, 5, and 14 following treatments with both CPT-FPV and sotrovimab. However, hs-CRP levels showed a greater decrease in the sotrovimab group between days 2–5 and 0–14 (Fig. S6).

## Immunologic response

The CPT-FPV group demonstrated a significant increase in nucleocapsid IgG antibody levels between days 0–2 ($P < 0.001$) and days 0–5 ($P < 0.001$), while there was a greater increase in the sotrovimab group between days 5–14 ($P = 0.023$). Higher levels of nucleocapsid IgG antibodies were detected in the CPT-FPV group on days 2 and 5 ($P < 0.01$, $P = 0.023$). Nucleocapsid IgG was rapidly decreased on day 28, with no difference in both groups. The surrogate virus neutralizing test (sVNT) revealed more positive neutralizing antibodies in the CPT-FPV group on days 5 and 14 ($P = 0.013$, $P = 0.028$). In contrast, the sotrovimab group had higher levels of anti-RBD spike IgG on days 2, 5, and 14 ($P < 0.001$, $P < 0.001$, and $P = 0.041$) and greater increase in anti-RBD spike IgG levels between days 2–5 and 5–14. However, on day 28, there was no significant difference ($P = 0.065$). On day 2, 100% of the sotrovimab group tested positive for anti-RBD spike IgG, compared to 89.7% in the CPT-FPV group (Fig. 3; Table S8).

## Immunomodulatory effects

No significant differences were observed in the levels of inflammatory cytokines on days 0, 2, and 5 between the two treatment groups. However, interleukin-6 (IL-6) levels showed a greater decrease in the sotrovimab group between days 2–5 and 0–14. On day 14, the CPT-FPV-treated group showed higher monocyte chemoattractant protein-1 (MCP-1) levels than sotrovimab (Table S9).

## Adverse events and safety

Two patients in the CPT group experienced anaphylaxis, a severe adverse event, during CCP transfusion and required hospitalization. They received short-term oxygen support and fully recovered within 24 hours. There was no notable difference in hepatotoxicity between the two groups. However, uric acid levels showed a significant increase in the CPT-FPV group between day 0 and day 5 ($P < 0.001$) but rapidly decreased to a similar level as the other group by day 14 (Tables S3 and S7).

## DISCUSSION

The multi-center, open-label, randomized control trial demonstrated that the combination of high neutralizing antibody titer CPT and FPV treatment for outpatient man-

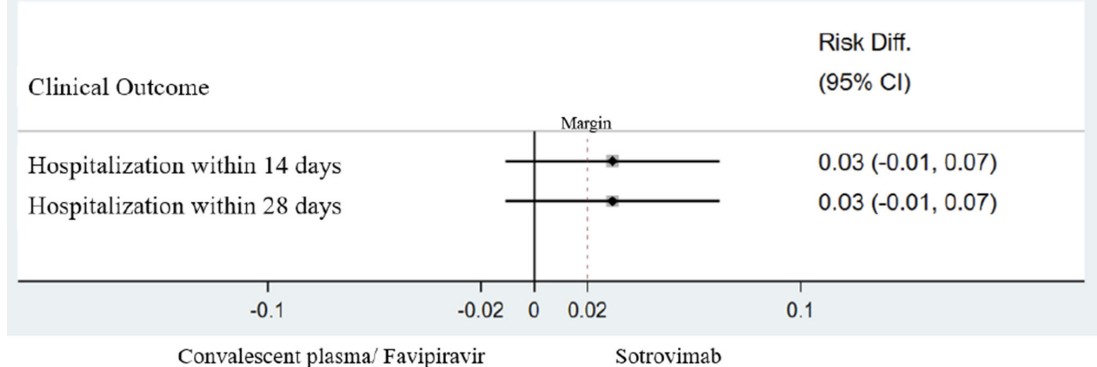

**FIG 2** The intention to treat analysis of non-inferiority of primary endpoints between COVID-19 convalescent plasma therapy/favipiravir and sotrovimab.

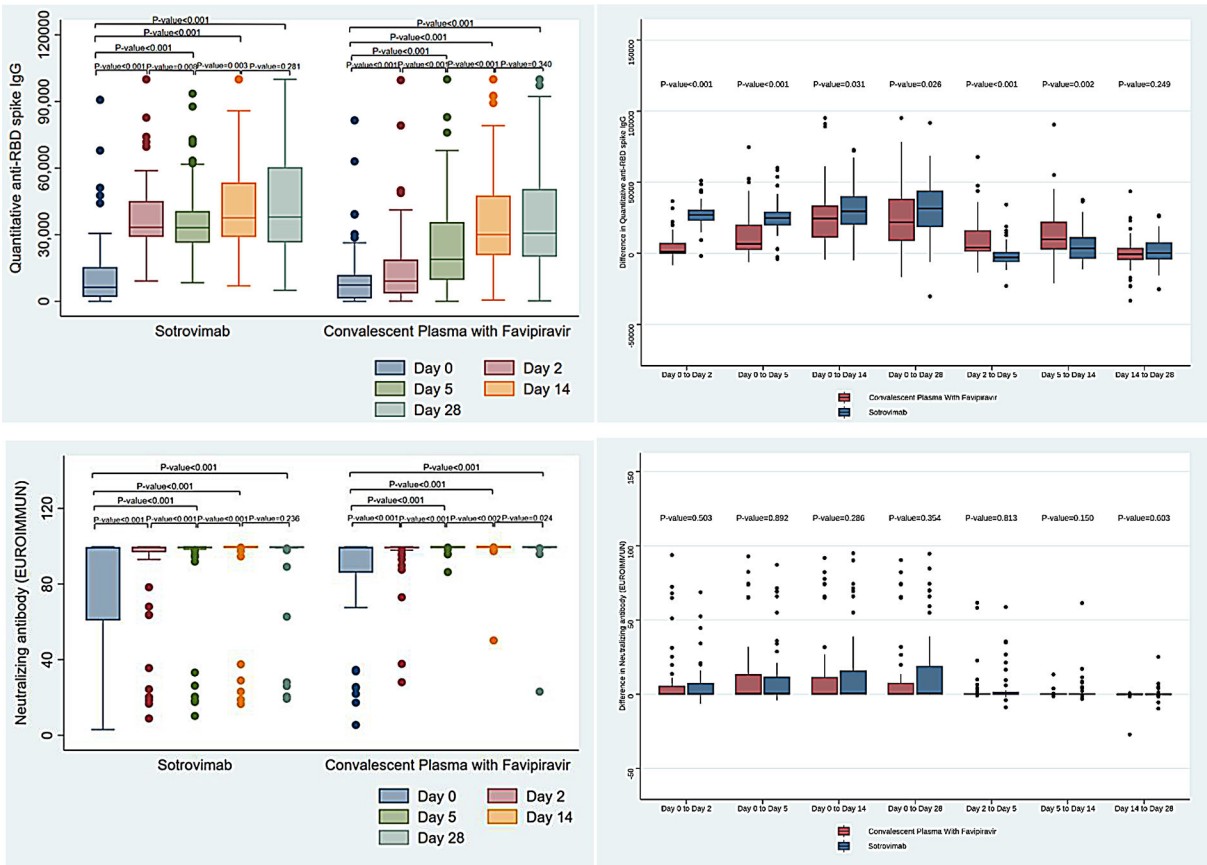

**FIG 3** Anti-RBD spike IgG and surrogated virus neutralizing test change after sotrovimab versus combined COVID-19 convalescent plasma with favipiravir treatment.

agement in mild-to-moderate COVID-19 with high-risk comorbidities was non-inferior to sotrovimab in preventing hospitalization within 14 and 28 days. This study may stand unique in comparing a monoclonal antibody to CCP, thereby presenting a novel contribution to the field.

Although two patients in the CPT-FPV group required hospitalization and short-term oxygen supplementation due to plasma-related anaphylaxis and complete recovery within 24 hours, no hospitalizations occurred due to severe COVID-19 in either treatment group. The incidence of serious adverse events did not differ significantly from the sotrovimab group. No cases of transfusion-related acute lung injury or transfusion-associated circulatory overload were observed in the CPT group.

Additionally, there were no significant differences in secondary outcomes, including emergency department visits, ICU admission rates, oxygen supplementation, invasive mechanical ventilation, and mortality. Although our study found that the CPT-FPV group had more increased infiltration evaluated by the CT severity index score on day 5, it was not clinically significant to develop severe COVID-19 that contributed to hospitalization, oxygen supplementation, or death. Although pneumonia was detected via chest CT on day 5 in 41% of individuals receiving CPT-FPV and 38% of those administered sotrovimab, a subgroup analysis of patients diagnosed with pneumonia on day 5 in both treatment groups showed no significant differences in several key indicators: clinical improvement or deterioration, oxygen saturation, WHO-CPS score, rate of pneumonia progression, Ct value, and cytokine levels (Table 3; Table S12).

Our study demonstrated non-inferiority in early intervention, within an average of 2 days after symptom onset, using high neutralizing antibody titer CCP infusion (>1:320) in high-risk patients, including a significant proportion of patients aged 65 or older, in an

**TABLE 2** Intention to treat analysis comparing the primary and secondary clinical outcomes of mild-to-moderate COVID-19 patients treated with convalescent plasma/favipiravir versus those treated with sotrovimab[a,b,c]

| Parameters | COVID-19 convalescent plasma/favipiravir (*n* = 68) | Sotrovimab (*n* = 68) | Difference in proportions (95% CI) | *P*-value |
|---|---|---|---|---|
| Hospitalization in 14 days, *n* (%) | 2 (2.94) | 0 (0.00) | 0.03 (−0.01, 0.07) | 0.4962 |
| Hospitalization in 28 days, *n* (%) | 2 (2.94) | 0 (0.00) | 0.03 (−0.01, 0.07) | 0.4962 |
| Hospitalization from severe COVID-19 in 14 days, *n* (%) | 0 (0.00) | 0 (0.00) | 0.00 (0.00, 0.00) | 1.0002 |
| Hospitalization from severe COVID-19 in 28 days, *n* (%) | 0 (0.00) | 0 (0.00) | 0.00 (0.00, 0.00) | 1.0002 |
| Emergency department visit in 14 days, *n* (%) | 2 (2.94) | 0 (0.00) | 0.03 (−0.01, 0.07) | 0.496[e] |
| Emergency department visit in 28 days, *n* (%) | 2 (2.94) | 0 (0.00) | 0.03 (−0.01, 0.07) | 0.496[e] |
| ICU admission in 28 days, *n* (%) | 0 (0.00) | 0 (0.00) | 0.00 (0.00, 0.00) | 1.000[e] |
| Death in 28 days, *n* (%) | 0 (0.00) | 0 (0.00) | 0.00 (0.00, 0.00) | 1.000[e] |
| Oxygen supplementation, *n* (%) | 2 (2.94) | 0 (0.00) | 0.03 (−0.01, 0.07) | 0.496[e] |
| Oxygen cannula, *n* (%) | 2 (2.94) | 0 (0.00) | 0.03 (−0.01, 0.07) | 0.496[e] |
| HFNC, *n* (%) | 0 (0.00) | 0 (0.00) | 0.00 (0.00, 0.00) | 1.000[e] |
| Invasive mechanical ventilation, *n* (%) | 0 (0.00) | 0 (0.00) | 0.00 (0.00, 0.00) | 1.000[e] |
| Clinical day 2 | | | | 0.654[d] |
| Stable | 15 (22.06) | 19 (27.94) | −0.06 (−0.20, 0.09) | |
| Worsening | 7 (10.29) | 5 (7.35) | 0.03 (−0.07, 0.12) | |
| Improvement | 46 (67.65) | 44 (64.71) | 0.03 (−0.13, 0.19) | |
| Clinical day 5 | | | | 0.687[e] |
| Stable | 2 (2.94) | 4 (5.88) | −0.03 (−0.10, 0.04) | |
| Worsening | 3 (4.41) | 4 (5.88) | −0.01 (−0.09, 0.06) | |
| Improvement | 63 (92.65) | 60 (88.24) | 0.04 (−0.05, 0.14) | |
| Clinical day 14 | | | | 0.240[e] |
| Stable | 2 (2.94) | 0 (0.00) | 0.03 (−0.01, 0.07) | |
| Worsening | 3 (4.41) | 1 (1.47) | 0.03 (−0.03, 0.09) | |
| Improvement | 63 (92.65) | 67 (98.53) | −0.06 (−0.13, 0.01) | |
| WHO clinical progression scale day 5, *n* (%) | | | | 0.571[e] |
| 0 | 4 (5.88) | 7 (10.29) | −0.04 (−0.14, 0.05) | |
| 1 | 58 (85.29) | 54 (79.41) | 0.06 (−0.07, 0.19) | |
| 2 | 6 (8.82) | 7 (10.29) | −0.01 (−0.11, 0.08) | |
| WHO clinical progression scale day 14, *n* (%) | | | | 0.099[e] |
| 0 | 52 (76.47) | 60 (88.24) | −0.12 (−0.24, 0.01) | |
| 1 | 14 (20.59) | 8 (11.76) | 0.09 (−0.03, 0.21) | |
| 2 | 2 (2.94) | 0 (0.00) | 0.03 (−0.01, 0.07) | |

[a]Data are presented as frequency and percentage for each parameter. Differences in proportions between the two treatment groups are provided with their 95% confidence interval.
[b]*n*: number; ICU: intensive care unit; HFNC: high-flow nasal cannula; WHO: World Health Organization; COVID-19: coronavirus disease 2019; CI: confidence interval.
[c]*P*-value: significance value, with values less than 0.05 typically indicating statistical significance.
[d]Calculations were done using the Pearson chi-squared test or.
[e]Fisher's exact test where appropriate.

outpatient setting. However, the observed effects could be influenced by the combined impact of FPV and CCP therapy.

Compared to the CSSC-004 study, CPT showed an absolute risk reduction of 3.4 percent points for hospitalization; both studies involved the transfusion of CCP with a titer of more than 1:320 in an outpatient setting to reduce COVID-19-related hospitalization. Both studies administered CCP within 7 days, 2 days in our study, whereas 6 days in CSSC-004. Notably, the CSSC-004 study occurred during the surge of the B.1.1.7 and B.1.617.2 variants, with over 80% of participants being unvaccinated and of the white race. Only 7% of participants in each arm were aged over 65 years. In contrast, our study revealed that over 65% of participants were over 65 years old and about 35% of omicron BA.1, BA.1.1, and BA.2 despite 80% being vaccinated (42).

Despite the high vaccination rate observed in our study, the PANORAMICS trial provided evidence of the effectiveness of early antiviral treatment with molnupiravir. This treatment demonstrated a reduced recovery time and alleviation of symptoms

**TABLE 3** Analysis of radiologic and virological outcomes in COVID-19 patients treated with sotrovimab versus combined COVID-19 convalescent plasma therapy with favipiravir[a,b,c]

| Parameters | COVID-19 convalescent plasma/favipiravir (n = 68) | Sotrovimab (n = 68) | Difference in proportions (95% CI) | P-value |
|---|---|---|---|---|
| **Radiological outcomes** | | | | |
| Pneumonia diagnosis at day 5, n (%) | 28 (41.18) | 26 (38.24) | 0.03 (−0.13, 0.19) | 0.726[d] |
| Radiologic change by chest CT on day 5, n (%) | | | | 0.258[e] |
| No pneumonia | 38 (55.88) | 42 (61.76) | −0.06 (−0.22, 0.11) | |
| Stable pneumonia | 19 (27.94) | 22 (32.35) | −0.04 (−0.20, 0.11) | |
| Progression | 9 (13.24) | 4 (5.88) | 0.07 (−0.02, 0.17) | |
| Resolution | 2 (2.94) | 0 (0.00) | 0.03 (−0.01, 0.07) | |
| **Virological outcomes** | | | | |
| Negative SARS-CoV-2 PCR on day 5, n (%) | 5 (7.35) | 9 (13.43) | −0.06 (−0.16, 0.04) | 0.273[d] |
| Negative SARS-CoV-2 PCR on day 14, n (%) | 57 (83.82) | 62 (91.18) | −0.07 (−0.18, 0.04) | 0.195[d] |

[a]Outcomes are presented in terms of frequency and percentage for each parameter. Proportional differences between the two treatment modalities are highlighted along with their corresponding 95% confidence intervals.
[b]n: number; CT: computed tomography; PCR: polymerase chain reaction; SARS-CoV 2: severe acute respiratory syndrome coronavirus-2; COVID-19: coronavirus disease 2019; CI: confidence intervals.
[c]P-value: significance value, with values less than 0.05 typically indicating statistical significance.
[d]Calculations were done using the Pearson chi-squared test or.
[e]Fisher's exact test where appropriate.

for individuals with 90% vaccination coverage, 69% high-risk comorbidities, and over 65 years of age compared to standard care. However, no significant differences were observed regarding hospitalization or death during the omicron surge (43). Consistent findings across studies targeting individuals aged 50 and 65 years and above indicate that the administration of nirmatrelvir/ritonavir resulted in lower rates of hospitalization and death compared to those who did not receive the treatment, irrespective of their prior immunization status (44, 45).

The CCP-Argentina study showed that CCP reduced the risk of severe respiratory disease by 16%, compared to a 31% risk reduction in the placebo group, which focused on patients over 65 years old, with 86% having at least one coexisting condition. These patients received a median neutralizing antibody titer of 1:3,200 CCP within 3 days. Recent studies and our data further support the beneficial effects of CCP treatment in elderly patients (33).

In contrast, the CoV-Early trial demonstrated that CCP failed to improve the 5-point disease severity scale among individuals aged 18–65 with high-risk comorbidities during the dominance of D614G and B.1.1.7 variants. Forty percent of participants received CCP more than 5 days after onset. The trial had lower rates of full vaccination and positive immunity than our study. Furthermore, the trial's discontinuation of 421 out of 690 participants from the almost fully vaccinated population affected its statistical power to detect large effect sizes, potentially influencing the outcomes. The trial also utilized a lower neutralizing antibody titer, with 55% of participants receiving CCP with a titer less than 386. These characteristics may have influenced the CCP outcomes (46). The SIREN-C3PO study focused on high-risk patients with mild symptomatic COVID-19, with approximately 60% of the participants aged over 50 years, 59% classified as obese, and 12.8% being immune-suppressed individuals in the CCP group. These patients received 250 mL of CCP with a median titer of 1:641, and no significant difference was found in the proportion of disease progression (35).

The REMAP-CAP, CONCOR-1, RECOVERY, PLACID, and RECOVER trials failed to show any clinical benefit of CCP in severe COVID-19 patients (20–22, 24, 25). Our study supports CPT with a high neutralizing titer in treating mild-to-moderate COVID-19 cases in outpatient settings non-inferior to sotrovimab.

Moreover, our study illustrated that no substantial differences were discerned in mitigating viral shedding, inflammatory markers, and cytokine alterations between the CPT and sotrovimab-treated groups. Nevertheless, the sotrovimab group manifested a more pronounced reduction in high-sensitivity C-reactive protein and interleukin-6, as

well as MCP-1 levels between days 2–5 and 0–14. While no discrepancy was observed in endpoint cytokine levels on consecutive days subsequent to CPT-FPV and sotrovimab treatment, this discovery endorses the perspective that NmaB may effectively preclude the evolution of severe COVID-19 and cytokine storms

However, the observation of a larger decline in IL-6 among sotrovimab recipients between days 5 and 14 is nuanced by the range of measurements, complicating its evaluation. Furthermore, the anti-inflammatory activity of IL-6 warrants attention, especially considering the elevated IL-10 levels among CPT-FPV recipients, even though they span a wide measurement range. Noteworthily, the anti-inflammatory cytokine, TGF-β, can be induced by MCP-1, which was observed to be higher in CPT-FPV recipients. These findings give rise to the hypothesis that CPT-FPV may modulate the inflammatory response toward an anti-inflammatory profile, potentially mediated by nucleocapsid antibodies (47, 48). These cytokine findings might elucidate previous results regarding cytokine levels, albeit with the caveat that plasma cytokine concentrations may not accurately mirror those within tissues.

The impacts of CPT-FPV and sotrovimab on the immune response may diverge. While participants treated with CPT-FPV exhibited elevated levels of SARS-CoV-2 nucleocapsid IgG on days 2 and 5, those receiving sotrovimab demonstrated higher levels of anti-spike RBD IgG on days 2, 5, and 14 and potentially day 28, likely attributable to sotrovimab's extended half-life. The post-administration elevation of antibody titers in participants who received sotrovimab is not unexpected, considering that the mass of antibodies delivered through monoclonal doses is typically greater than that found in CCP. Nonetheless, it is imperative to acknowledge that CCP, being a polyclonal reagent, offers diversity in the recognition of epitopes and isotype functionality. Nevertheless, recipients of CPT-FPV demonstrated higher sVNT and positive neutralizing antibodies on days 2, 5, and 14. Although these disparities were not observed on day 28, they imply that CPT-FPV may have exerted more pronounced early antiviral activity, potentially due to its inclusion of isotypes other than IgG and additional SARS-CoV-2 specificities. The SARS-CoV-2 nucleocapsid-specific antibodies derived from CCP contained a higher quantity of functional antibodies than recipients' plasma, and these functions are associated with attenuated humoral immune evolution in patients exhibiting high-titer anti-spike IgG (47, 48). This consideration becomes pertinent as viral evolution has induced SARS-CoV-2 resistance to all RBD monoclonals, thereby facilitating the selection of resistant variants.

Sotrovimab, NmaB therapy, had a well-known benefit from marked higher antibody content than CCP and improved the outcome from early outpatient treatment. However, the resistant variant virus was the main concern. While the efficacy difference between CCP and NmaB mainly revolves around the dose gap, this study specifically selected convalescent plasma with a titer above 320 to minimize the difference. Nonetheless, there is ongoing controversy regarding the sufficient titer of antibodies needed to combat each variant of concern (49). Based on the trial, it was established that 30–40% of virus isolates in both groups were identified as the omicron variant. The trial spanned from January to 23 April 2022, aligning with the tail end of the B.1.617.2 wave and almost completely overlapping with the B.1.1.529, BA.* wave (50) (Fig. S14). There exists a plausible scenario wherein the CCP, derived from infections associated with the alpha and delta variants, might exhibit reduced neutralizing titers and limited utility of antibodies against the circulating omicron variants. Furthermore, an *in vitro* study of sotrovimab revealed a less than two fold decrease in neutralizing activity against omicron, raising concerns regarding the potentially compromised efficacy of sotrovimab against this variant (17). Nevertheless, recent studies conducted during the BA.1 and BA.2 periods have demonstrated that sotrovimab is associated with reduced hospitalization and mortality (16, 51–53).

Moreover, our study did not explore using VaxPlasma (hybrid plasma) obtained from vaccinated donors, which contains antibodies with 10 times higher titers than standard high-titer CCP and has shown high effectiveness against the omicron variant.

Although some CCP donors in our study may have received inactivated or ChAdOx-1 nCoV vaccinations, we could not clarify their impact (42, 54–56).

Our study suggests that a titer of CCP higher than 1:320 may be sufficient to demonstrate non-inferiority to sotrovimab. In our immunologic study, we observed that sotrovimab increased RBD spike antibodies from day 0 to 28, but there was no significant difference in antibody levels after day 14. Although there was no significant difference in sVNT changes between the two treatments at any interval after infusion, the sVNT in CCP was higher than in sotrovimab on day 5 (Fig. 3; Table S8).

Our study had several limitations that need to be acknowledged. Firstly, there was a relatively low percentage of unvaccinated individuals in both CPT-FPV and sotrovimab groups (15% and 25%, respectively), which may have influenced the observed clinical outcomes and introduced a bias toward more favorable results. Secondly, the lack of information regarding the immunization status of the plasma donors including vaccinated donors may have impacted the interpretation of clinical outcomes associated with CPT. Thirdly, all patients in the CPT group received FPV following the Thai COVID-19 standard treatment guidelines, potentially confounding the outcomes due to the antiviral activity of the drug, although the efficacy of FPV in Asians remained controversial. Fourthly, the sVNT used in the study could not detect neutralizing antibodies against the delta variants and CCP from individuals infected during the surges of the alpha and delta variants. Fifth, we are unable to provide the actual level of neutralizing antibody, the isotypes of nucleocapsid, and/or spike and/or RBD antibodies titer in each bag; we can only confirm that the CCP possessed a neutralizing antibody level above 1:320, as prepared by the Thai National Blood Center. Lastly, while 30% of patients were infected with the omicron variant, most patients in both treatment arms (60%) were infected with unknown variants, which may impact the generalizability of the findings.

## Conclusions

High neutralizing titer CCP therapy combined with FPV showed non-inferiority to sotrovimab in preventing hospitalization, emergency room visits, oxygen supplementation, and mortality in mild-to-moderate COVID-19 cases. This treatment approach may serve as an alternative to neutralizing monoclonal antibodies when dealing with emerging variants that have spike protein mutations.

## ACKNOWLEDGMENTS

We thank all healthcare providers who work in the frontline at the Acute Respiratory Tract Infection clinic, COVID-19 outpatient clinic, and cohort wards for supporting and providing this study's clinical data. All authors thank and extend the deepest sympathies and condolences to the victims and their families. We thank the Center of Excellence in Clinical Virology, Department of Pediatrics, Faculty of Medicine, Chulalongkorn University Bangkok, Thailand, for the expert consultant and National Blood Center for providing COVID-19 convalescent plasma in the study.

This work was supported by Chulabhorn Royal Academy, Bangkok, Thailand.

T.S. had full access to all of the data in this study and took responsibility for the data's integrity and accuracy. T.S. and N.M. performed the concept and design. T.S., N.N., K.T., O.S., K.S., Y.P., and N.M. performed the investigation. T.S. performed the acquisition, analysis, or interpretation of data. T.S. performed the critical revision of the manuscript for important intellectual content. K.S. performed the statistical analysis. N.M. and T.S. obtained the funding. T.S., N.N., K.T., O.S., K.S., Y.P., and N.M. provided administrative, technical, or material support. T.S. and N.M. performed the supervision.

## AUTHOR AFFILIATIONS

[1]Division of Infectious Diseases, Department of Medicine, Chulabhorn Hospital, Chulabhorn Royal Academy, Bangkok, Thailand

[2]Princess Srisavangavadhana College of Medicine, Chulabhorn Royal Academy, Bangkok, Thailand

[3]Department of Medicine, Prachathiput Hospital, Pathum Thani, Thailand

[4]Center of Learning and Research in Celebration of HRH Princess Chulabhorn 60th Birthday Anniversary, Chulabhorn Royal Academy, Bangkok, Thailand

[5]Department of Pediatrics, Center of Excellence in Clinical Virology, Faculty of Medicine, Chulalongkorn University, Bangkok, Thailand

[6]Chulabhorn Hospital, Chulabhorn Royal Academy, Bangkok, Thailand

## AUTHOR ORCIDs

Taweegrit Siripongboonsitti ⓘ http://orcid.org/0000-0001-7256-9982

## FUNDING

| Funder | Grant(s) | Author(s) |
| --- | --- | --- |
| Chulabhorn Royal Academy (CRA) | | Taweegrit Siripongboonsitti |

## AUTHOR CONTRIBUTIONS

Taweegrit Siripongboonsitti, Conceptualization, Data curation, Formal analysis, Funding acquisition, Investigation, Methodology, Project administration, Resources, Software, Supervision, Validation, Visualization, Writing – original draft, Writing – review and editing | Nuttakant Nontawong, Investigation, Resources | Kriangkrai Tawinprai, Investigation | Ornpreya Suptawiwat, Investigation, Resources | Kamonwan Soonklang, Data curation, Formal analysis, Investigation, Software, Validation, Visualization | Yong Poovorawan, Supervision | Nithi Mahanonda, Supervision

## DATA AVAILABILITY

The datasets generated and analyzed during the current study are not publicly available due to avoiding use for wrong purposes but are available from the corresponding author on reasonable request.

## ETHICS APPROVAL

This study was approved by the Ethics Committee for Human Research, Chulabhorn Research Institute (EC No. 192/2564). All severe COVID-19 patients were informed about the treatment, and all patients had written the consent forms before admission to receive all treatment options that might benefit all severe COVID-19 patients in addition to a standard of care during the outbreak. Specific CCP consent was not applicable. Our study followed the Helsinki Declaration of 1964 and its later amendments. Consent for publication was not applicable.

## ADDITIONAL FILES

The following material is available online.

### Supplemental Material

**Supplemental file 1 (Spectrum03257-23-s0001.pdf).** Supplemental material.

### Open Peer Review

**PEER REVIEW HISTORY (review-history.pdf).** An accounting of the reviewer comments and feedback.

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
