## [Reviewer comments · Microbiology Spectrum]

Microbiology Spectrum

Efficacy of Combined COVID-19 Convalescent Plasma with Oral RNA-dependent RNA Polymerase Inhibitor Treatment versus Monoclonal Neutralizing Antibody Therapy in COVID-19 Outpatients: Multicenter, Non-inferiority, Open-label Randomized Controlled Trial (PlasMab)

Taweegrit Siripongboonsitti, Nuttakant Nontawong, Kriangkrai Tawinprai, Ornpreeya Suptawiwat, Kamonwan Soonklang, Yong Poovorawan, and Nithi Mahanonda

Corresponding Author(s): Taweegrit Siripongboonsitti, Chulabhorn Royal Academy

Review Timeline:

Submission Date:	September 1, 2023
Editorial Decision:	October 7, 2023
Revision Received:	October 17, 2023
Accepted:	October 26, 2023

Editor: Oliver Laeyendecker

Reviewer(s): The reviewers have opted to remain anonymous.

Transaction Report:

DOI: <https://doi.org/10.1128/spectrum.03257-23>

October 7, 2023

Dr. Taweegrit Siripongboonsitti
Chulabhorn Royal Academy
Medicine
906 Kamphaeng Phet 6 Rd.
Talad Bang Khen
Laksi, Bangkok 10210
Thailand

Re: Spectrum03257-23 (Efficacy of Combined COVID-19 Convalescent Plasma with Oral RNA-dependent RNA Polymerase Inhibitor Treatment versus Monoclonal Neutralizing Antibody Therapy in COVID-19 Outpatients: Multicenter, Non-inferiority, Open-label Randomized Controlled Trial (PlasMab))

Dear Dr. Taweegrit Siripongboonsitti:

Thank you for submitting your manuscript to Microbiology Spectrum. As you will see your paper is very close to acceptance. Please modify the manuscript along the lines I have recommended. As these revisions are quite minor, I expect that you should be able to turn in the revised paper in less than 30 days, if not sooner. If your manuscript was reviewed, you will find the reviewers' comments below.

When submitting the revised version of your paper, please provide (1) point-by-point responses to the issues raised by the reviewers as file type "Response to Reviewers," not in your cover letter, and (2) a PDF file that indicates the changes from the original submission (by highlighting or underlining the changes) as file type "Marked Up Manuscript - For Review Only". Please use this link to submit your revised manuscript. Detailed instructions on submitting your revised paper are below.

Link Not Available

Sincerely,

Oliver Laeyendecker

Reviewer comments:

Reviewer #1 (Comments for the Author):

This paper reports results of a randomized multi-center, non-inferiority, open label, randomized controlled trial conducted in Thailand comparing the efficacy of high titer COVID-19 convalescent plasma (CCP) and favipiravir with sotrovimab in preventing hospitalization outpatients with mild to moderate COVID-19. The results show no difference between groups, which is important for the use of CCP since sotrovimab has established efficacy in this indication. CCP is a relatively inexpensive therapy compared with monoclonal antibodies that is available anywhere in the world where there are transfusion services and people who have recovered from COVID-19. Hence, the results of this study provide additional evidence for the efficacy of CCP and hopefully will encourage more clinical investigation. Unfortunately, limitations of this study ranging from changes in the prevalence of SARS-CoV-2 variants and a high baseline prevalence of antibodies in the participant population diminish the solidity of any conclusions from this work. This reviewer has no major criticisms of the work done but has suggestions for improvement.

1. The paper would benefit from text about the rationale for this clinical trial. Why combine favipiravir with CCP? The discussion hints that this was because of Thai guidelines but this is not stated clearly. A citation to the Thai guidelines would be helpful. The

introduction is primarily about CCP and says nothing about favipiravir. While the efficacy of favipiravir remains uncertain, there is some evidence also from Thailand that favipiravir is effective in reducing progression of disease (PMID: 37689754). Hence, we don't know if the effective therapeutic was CCP, favipiravir or the combination of CCP-favipiravir.

2. The trial was carried out from January 22, 2022, to April 23, 2022. This was in the early omicron wave. The CCP dated from infections with the earlier variants and sotrovimab lost efficacy against omicron. Table 1 shows that between 30-40% of virus isolates in both groups were omicron, for which sotrovimab therapy was ineffective and CCP from earlier variants would have lower neutralizing titers. Although this limitation is implied in the discussion perhaps it can be stated more clearly.

3. The higher antibody titers in participants after administration of sotrovimab are not surprising since the mass of antibody administered with monoclonal doses is greater than that in CCP, with the caveat that CCP is a polyclonal reagent that provides diversity in epitopes recognize and isotype function. The authors may want to state this.

4. This study may be the only one that has compared a monoclonal antibody to CCP. This gives it novelty.

5. The discussion is repetitive and could easily be consolidated. For example, the IL-6 results are repeated three times in different paragraphs. Please tighten the discussion.

Reviewer #2 (Comments for the Author):

In this article, Siripongboonsitti and colleagues report the results of their outpatient non-inferiority randomized trial of the efficacy of the combination of COVID convalescent plasma therapy (CPT) with oral favipiravir (FPV) versus sotrovimab. The study was conducted in Thailand from January to April 2022. The results showed that CPT-FPV (n=68) was non-inferior to sotrovimab (n=68) for the primary outcome, hospitalization within 14 days and 28 days of treatment. In addition, there were no differences between the treatments for emergency room visits or other clinical outcomes that were evaluated (e.g., oxygen supplementation, need for mechanical ventilation, or mortality). Notably, the authors report that ~30% of each treatment group had pneumonia at the time of treatment.

The authors should be congratulated for conducting this well-designed study. The results are important and compelling, with the caveat that the role that PPV plays in the activity of CPT requires further investigation. First, the demonstration that CPT-FPV is non-inferior to sotrovimab in this well-done randomized trial should encourage larger non-inferiority trials with monoclonal antibodies, should they become available. However, given the likelihood that such trials will not be conducted, the results from this study should compel public health entities to prioritize CPT for early COVID-19, especially in settings where high priced therapies are not available. The non-inferiority of CPT-FPV in this study reinforces the large body of evidence that CPT efficacy against COVID-19 requires early administration. Study participants had a mean symptom duration of <3 days, >65% in each group were older than 60yrs, very few had prior COVID-19, 85% (CPT-FPV) and 75% (sotrovimab) were vaccinated, 79% (CPT-FPV) and 90% (sotrovimab) were seropositive for RBD IgG at baseline, and ~30% in each group had pneumonia at baseline.

The authors present very comprehensive clinical and immunological data that provides an in-depth picture of the effect of each treatment on the primary and exploratory outcomes. However, the measure of central tendency reported (mean, median), confidence intervals, and statement of what is in parentheses should be included in table legends. In addition, the following observations are offered for the authors' consideration as they may provide insight into the benefit of CPT-FPV in their study and strengthen the case for future use of COVID convalescent plasma to treat early COVID-19.

1. The effect of CPT-FPV and sotrovimab on the immune response may differ. Whereas CPT-FPV-treated participants had higher levels of CLIA SARS-CoV-2 (nucleocapsid) IgG on days 2 and 5, sotrovimab recipients had higher levels of RBD IgG on days 2, 5, and 14, and possibly day 28, likely due to the long half-life of sotrovimab. Nonetheless, CPT-FPV recipients had higher sVNT and 'positive neutralizing antibody' on days 2, 5, and 14. While not observed on day 28, these differences suggest that CPT-FPV may have had more early anti-viral activity, perhaps because it included isotypes other than IgG and additional SARS-CoV-2 specificities. This is important since viral evolution resulted in SARS-CoV-2 resistance to all RBD monoclonals, which contributed to selection of resistant variants. Are the following known?

The neutralizing titer of each CCP unit that was administered? The nucleocapsid antibody titer? The isotypes of nucleocapsid and/or spike and/or RBD antibodies? The time of CCP collection relative to when each unit was given.

If available, this information would be helpful in establishing guidelines for CCP use, particularly with the circulating (variant) viral strain is not known.

2. Thirty percent of each treatment group had pneumonia at baseline and 30% had radiological evidence of pneumonia 5 days after treatment. Are participant oxygen saturation measurements available? Other evidence of tissue damage stemming from pneumonia? Would CT scans have been done if the participants were not enrolled in the trial?

3. The cytokine data is interesting. The authors note a larger decline in IL-6 in sotrovimab recipients between days 5 and 14, but the range of measurements makes this difficult to evaluate. Moreover, IL-6 can have anti-inflammatory activity, which is notable given the higher IL-10 level among CPT-FPV recipients, albeit with a wide range of measurements. Notably, the anti-inflammatory cytokine TGF-beta can be induced by MCP-1, which was higher in CPT-FPV recipients. These findings suggest the hypothesis that CPT-FPV may skew the inflammatory response towards an anti-inflammatory profile, perhaps mediated by nucleocapsid antibodies (see Herman et al Nat Commun 2021 and Cell Rep Med 2022). More on the characteristics of the CPT

(see point 1) may shed light on the cytokine findings, with the caveat that plasma cytokine levels may not reflect tissue levels.

Preparing Revision Guidelines

Please return the manuscript within 60 days; if you cannot complete the modification within this time period, please contact me. If you do not wish to modify the manuscript and prefer to submit it to another journal, please notify me of your decision immediately so that the manuscript may be formally withdrawn from consideration by Microbiology Spectrum.

Response to reviewer

Reviewer #1 (Comments for the Author):

This paper reports results of a randomized multi-center, non-inferiority, open label, randomized controlled trial conducted in Thailand comparing the efficacy of high titer COVID-19 convalescent plasma (CCP) and favipiravir with sotrovimab in preventing hospitalization outpatients with mild to moderate COVID-19. The results show no difference between groups, which is important for the use of CCP since sotrovimab has established efficacy in this indication. CCP is a relatively inexpensive therapy compared with monoclonal antibodies that is available anywhere in the world where there are transfusion services and people who have recovered from COVID-19. Hence, the results of this study provide additional evidence for the efficacy of CCP and hopefully will encourage more clinical investigation. Unfortunately, limitations of this study ranging from changes in the prevalence of SARS-CoV-2 variants and a high baseline prevalence of antibodies in the participant population diminish the solidity of any conclusions from this work. This reviewer has no major criticisms of the work done but has suggestions for improvement.

1. The paper would benefit from text about the rationale for this clinical trial. Why combine favipiravir with CCP? The discussion hints that this was because of Thai guidelines but this is not stated clearly. A citation to the Thai guidelines would be helpful. The introduction is primarily about CCP and says nothing about favipiravir. While the efficacy of favipiravir remains uncertain, there is some evidence also from Thailand that favipiravir is effective in reducing progression of disease (PMID: 37689754). Hence, we don't know if the effective therapeutic was CCP, favipiravir or the combination of CCP-favipiravir.

Response: We are truly grateful for your feedback and affirmations.

Your insights have been instrumental, and in response, we have thoroughly revised the manuscript to incorporate your valuable suggestions.

Line 52: In Thailand, amid the surge of the Delta and Omicron variants, favipiravir (FPV) has been extensively deployed as the standard treatment for mild to moderate cases of COVID-19, adhering to the Thai COVID-19 Clinical Practice guidelines. This approach was necessitated by the constrained availability of novel therapeutic options (4). A real-world study within Thailand, alongside a meta-analysis, underscored the advantages of FPV treatment, revealing notable clinical improvements, diminished rates of clinical deterioration, enhanced viral clearance, decreased dependency on supplemental oxygen therapy, and reduced mortality (5-7). However, a Randomized Controlled Trial (RCT) conducted among obese, Hispanic individuals failed to demonstrate the efficacy of FPV treatment in mild-to-moderate COVID-19 (8). Given its non-

linear pharmacokinetics (PK) and auto-inhibition properties, FPV's efficacy remains a topic of controversy and perpetuates a treatment gap, especially among certain ethnic groups characterized by overweight and obesity

2. The trial was carried out from January 22, 2022, to April 23, 2022. This was in the early omicron wave. The CCP dated from infections with the earlier variants and sotrovimab lost efficacy against omicron. Table 1 shows that between 30-40% of virus isolates in both groups were omicron, for which sotrovimab therapy was ineffective and CCP from earlier variants would have lower neutralizing titers. Although this limitation is implied in the discussion perhaps it can be stated more clearly.

Line 379: Based on the trial, it was established that 30-40% of virus isolates in both groups were identified as the Omicron variant. The trial spanned from January to April 23, 2022, aligning with the tail end of the B.1.617.2 wave and almost completely overlapping with the B.1.1.529 wave (48). (Supplementary Figure S14) There exists a plausible scenario wherein the CCP, derived from infections associated with the Alpha and Delta variants, might exhibit reduced neutralizing titers and limited utility of antibodies against the circulating Omicron variants. Furthermore, sotrovimab might display compromised efficacy against Omicron although the in vitro study showed less than 2-fold decrease of neutralizing activity against omicron (16).

3. The higher antibody titers in participants after administration of sotrovimab are not surprising since the mass of antibody administered with monoclonal doses is greater than that in CCP, with the caveat that CCP is a polyclonal reagent that provides diversity in epitopes recognize and isotype function. The authors may want to state this.

Line 361: The post-administration elevation of antibody titers in participants who received sotrovimab is not unexpected, considering that the mass of antibodies delivered through monoclonal doses is typically greater than that found in CCP. Nonetheless, it is imperative to acknowledge that CCP, being a polyclonal reagent, offers a diversity in the recognition of epitopes and isotype functionality

4. This study may be the only one that has compared a monoclonal antibody to CCP. This gives it novelty.

Line 278: This study may stand unique in comparing a monoclonal antibody to CCP, thereby presenting a novel contribution to the field.

5. The discussion is repetitive and could easily be consolidated. For example, the IL-6 results are repeated three times in different paragraphs. Please tighten the discussion.

Line 340: Moreover, our study illustrated that no substantial differences were discerned in mitigating viral shedding, inflammatory markers, and cytokine alterations between the CPT and sotrovimab-treated groups. Nevertheless, the sotrovimab group manifested a more pronounced reduction in high-sensitivity C-reactive protein (hs-CRP) and interleukin-6 (IL-6), as well as MCP-1 levels between days 2-5 and 0-14. While no discrepancy was observed in endpoint cytokine levels on consecutive days subsequent to CPT-FPV and sotrovimab treatment, this discovery endorses the perspective that NmaB may effectively preclude the evolution of severe COVID-19 and cytokine storms

Reviewer #2 (Comments for the Author):

In this article, Siripongboonsitti and colleagues report the results of their outpatient non-inferiority randomized trial of the efficacy of the combination of COVID convalescent plasma therapy (CPT) with oral favipiravir (FPV) versus sotrovimab. The study was conducted in Thailand from January to April 2022. The results showed that CPT-FPV (n=68) was non-inferior to sotrovimab (n=68) for the primary outcome, hospitalization within 14 days and 28 days of treatment. In addition, there were no differences between the treatments for emergency room visits or other clinical outcomes that were evaluated (e.g., oxygen supplementation, need for mechanical ventilation, or mortality). Notably, the authors report that ~30% of each treatment group had pneumonia at the time of treatment.

The authors should be congratulated for conducting this well-designed study. The results are important and compelling, with the caveat that the role that PPV plays in the activity of CPT requires further investigation. First, the demonstration that CPT-FPV is non-inferior to sotrovimab in this well-done randomized trial should encourage larger non-inferiority trials with monoclonal antibodies, should they become available. However, given the likelihood that such trials will not be conducted, the results from this study should compel public health entities to prioritize CPT for early COVID-19, especially in settings where high priced therapies are not available. The non-inferiority of CPT-FPV in this study reinforces the large body of evidence that CPT efficacy against COVID-19 requires early administration. Study participants had a mean symptom duration of <3 days, >65% in each group were older than 60yrs, very few had prior COVID-19, 85% (CPT-FPV) and 75% (sotrovimab) were vaccinated, 79% (CPT-FPV) and 90% (sotrovimab) were seropositive for RBD IgG at baseline, and ~30% in each group had pneumonia at baseline.

The authors present very comprehensive clinical and immunological data that provides an in-depth picture of the effect of each treatment on the primary and exploratory outcomes.

Thank you for recognizing the value of our research and for your constructive comments.

We acknowledge the importance of larger non-inferiority trials with monoclonal antibodies, as these would significantly enhance the robustness of our findings. Unfortunately, as you've noted, our study's scope was constrained by funding limitations, preventing us from pursuing this rigorous approach at the current stage.

However, the measure of central tendency reported (mean, median), confidence intervals, and statement of what is in parentheses should be included in table legends.

We greatly appreciate your constructive suggestion. Following your advice, we have promptly revised the table legend to ensure greater clarity and accuracy.

In addition, the following observations are offered for the authors' consideration as they may provide insight into the benefit of CPT-FPV in their study and strengthen the case for future use of COVID convalescent plasma to treat early COVID-19.

1. The effect of CPT-FPV and sotrovimab on the immune response may differ. Whereas CPT-FPV-treated participants had higher levels of CLIA SARS-CoV-2 (nucleocapsid) IgG on days 2 and 5, sotrovimab recipients had higher levels of RBD IgG on days 2, 5, and 14, and possibly day 28, likely due to the long half-life of sotrovimab. Nonetheless, CPT-FPV recipients had higher sVNT and 'positive neutralizing antibody' on days 2, 5, and 14. While not observed on day 28, these differences suggest that CPT-FPV may have had more early anti-viral activity, perhaps because it included isotypes other than IgG and additional SARS-CoV-2 specificities. This is important since viral evolution resulted in SARS-CoV-2 resistance to all RBD monoclonals, which contributed to selection of resistant variants.

Response: Thank you for your insightful comments.

We want to confirm that we have incorporated your suggestions into the manuscript.

Are the following known?

The neutralizing titer of each CCP unit that was administered? The nucleocapsid antibody titer? The isotypes of nucleocapsid and/or spike and/or RBD antibodies? The time of CCP collection relative to when each unit was given.

If available, this information would be helpful in establishing guidelines for CCP use, particularly with the circulating (variant) viral strain is not known.

Response: We appreciate your inquiry about the neutralizing antibody titers in the CCP bags used in our study. We assure you that each bag met the minimum potency threshold with a titer above 1:320, as verified by the Thai National Blood Center in Bangkok, Thailand. However, we regret that we're unable to furnish individual titer levels for each bag.

Our initial methodology did include a plan to conduct detailed assessments of the neutralizing titers and antibody isotypes for each CCP unit. However, due to unforeseen funding constraints, we had to adapt our approach. This constraint and its potential impact on our study's breadth are acknowledged in the limitations section of our paper.

We understand the significance of these details in the broader context of our findings and appreciate your understanding of the logistical challenges we faced.

Line 411: “Fifth, we are unable to provide the actual level of neutralizing antibody, The isotypes of nucleocapsid and/or spike and/or RBD antibodies titer in each CCP (Convalescent Plasma) bag; we can only confirm that the CCP possessed a neutralizing antibody level above 1:320, as prepared by the Thai National Blood Center.”

2. Thirty percent of each treatment group had pneumonia at baseline and 30% had radiological evidence of pneumonia 5 days after treatment.

Are participant oxygen saturation measurements available?

Other evidence of tissue damage stemming from pneumonia?

Response: Thank you for your insightful comments.

We confirm that a detailed subgroup analysis was conducted for individuals with pneumonia by day 5, with results showing no notable differences, as presented in Supplementary Table S12. This information has also been incorporated into the discussion section for comprehensive context.

We appreciate your attention to detail and welcome any further feedback you might have.

Line 291: Although pneumonia was detected via chest CT on day 5 in 41% of individuals receiving CPT-FPV and 38% of those administered sotrovimab, A subgroup analysis of patients diagnosed with pneumonia on day 5 in both treatment groups showed no significant differences in several key indicators: clinical improvement or deterioration, oxygen saturation, WHO-CPS score, rate of pneumonia progression, Ct-value, and cytokine levels. (*Supplementary Table S12*)

Would CT scans have been done if the participants were not enrolled in the trial?

Response: You are correct; the cases in question are part of a separate cohort. Your idea presents a compelling direction for further analysis, and we are committed to diligently exploring these aspects in our ongoing research efforts

3. The cytokine data is interesting. The authors note a larger decline in IL-6 in sotrovimab recipients between days 5 and 14, but the range of measurements makes this difficult to evaluate. Moreover, IL-6 can have anti-inflammatory activity, which is notable given the higher IL-10 level among CPT-FPV recipients, albeit with a wide range of measurements. Notably, the anti-inflammatory cytokine TGF-beta can be induced by MCP-1, which was higher in CPT-FPV recipients. These findings suggest the hypothesis that CPT-FPV may skew the inflammatory response towards an anti-inflammatory profile, perhaps mediated by

nucleocapsid antibodies (see Herman et al Nat Commun 2021 and Cell Rep Med 2022). More on the characteristics of the CPT (see point 1) may shed light on the cytokine findings, with the caveat that plasma cytokine levels may not reflect tissue levels.

Response: We express our sincere gratitude for your valuable comments and insights.

In response to your feedback, we have carefully revised our manuscript to include the suggested statement and pertinent references. Your contributions have undoubtedly enriched the content and rigor of our work.

Re: Spectrum03257-23R1 (Efficacy of Combined COVID-19 Convalescent Plasma with Oral RNA-dependent RNA Polymerase Inhibitor Treatment versus Monoclonal Neutralizing Antibody Therapy in COVID-19 Outpatients: Multicenter, Non-inferiority, Open-label Randomized Controlled Trial (PlasMab))

Dear Dr. Taweegrit Siripongboonsitti:

Your manuscript has been accepted, and I am forwarding it to the ASM production staff for publication. Your paper will first be checked to make sure all elements meet the technical requirements. ASM staff will contact you if anything needs to be revised before copyediting and production can begin. Otherwise, you will be notified when your proofs are ready to be viewed.

Sincerely,
Oliver Laeyendecker
Editor
Microbiology Spectrum